# Bi-Allelic *MARVELD2* Variant Identified with Exome Sequencing in a Consanguineous Multiplex Ghanaian Family Segregating Non-Syndromic Hearing Loss

**DOI:** 10.3390/ijms26073337

**Published:** 2025-04-03

**Authors:** Elvis Twumasi Aboagye, Samuel Mawuli Adadey, Leonardo Alves de Souza Rios, Kevin K. Esoh, Edmond Wonkam-Tingang, Lettilia Xhakaza, Carmen De Kock, Isabelle Schrauwen, Lucas Amenga-Etego, Dirk Lang, Gordon A. Awandare, Suzanne M. Leal, Shaheen Mowla, Ambroise Wonkam

**Affiliations:** 1Department of Pathology, Division of Human Genetics, Faculty of Health Sciences, University of Cape Town, Cape Town 7925, South Africa or etaboagye@ug.edu.gh (E.T.A.); samuel.adadey@uct.ac.za (S.M.A.); kesohku1@jh.edu (K.K.E.); wonkamedmond@yahoo.fr (E.W.-T.); lettilia.xhakaza@uct.ac.za (L.X.); carmen.dekock@uct.ac.za (C.D.K.); 2West African Centre for Cell Biology of Infectious Pathogens (WACCBIP), University of Ghana, Legon, Accra LG 54, Ghana; lamengaetego@ug.edu.gh (L.A.-E.); gawandare@ug.edu.gh (G.A.A.); 3Department of Pathology, Division of Haematology, Faculty of Health Sciences, University of Cape Town, Cape Town 7925, South Africa; leonardo.rios@uct.ac.za (L.A.d.S.R.); shaheen.mowla@uct.ac.za (S.M.); 4McKusick-Nathans Institute & Department of Genetic Medicine, Johns Hopkins University School of Medicine, Baltimore, MD 21205, USA; 5Department of Translational Neurosciences, University of Arizona College of Medicine Phoenix, Phoenix, AZ 85004, USA; isabelle.schrauwen@gmail.com; 6Department of Human Biology, Division of Cell Biology, Faculty of Health Sciences, University of Cape Town, Cape Town 7925, South Africa; dirk.lang@uct.ac.za; 7Center for Statistical Genetics, Gertrude H. Sergievsky Center, Department of Neurology, Columbia University Medical Centre, New York, NY 10032, USA; sml3@cumc.columbia.edu; 8Taub Institute, Columbia University Medical Center, New York, NY 10032, USA

**Keywords:** non-syndromic hearing loss, *MARVELD2*, consanguinity, whole-exome sequencing, Ghana, Africa

## Abstract

Genetic studies and phenotypic expansion of hearing loss (HL) for people living in Africa are greatly needed. We evaluated the clinical phenotypes of three affected siblings presenting non-syndromic (NS) HL and five unaffected members of a consanguineous Ghanaian family. Analysis of exome sequence data was performed for all affected and one unaffected family members. In-depth genetic and cellular characterization studies were performed to investigate biological significance of the implicated variant using bioinformatic tools and cell-based experimentation. Audiological examinations showed severe-to-profound, bilateral, symmetrical, and post-lingual onset. The whole-exome sequencing (WES) identified a homozygous frameshift variant: MARVEL domain containing 2 (*MARVELD2*):c.1058dup;p.(Val354Serfs*5) in all affected siblings. This frameshift variant leads to an early stop codon insertion and predicted to be targeted by nonsense medicated decay (mutant protein predicted to lack conserved C-terminal domain if translated). Cell immunofluorescence and immunocytochemistry studies exposed the functional impact of the mutant protein’s expression, stability, localization, protein–protein binding, barrier function, and actin cytoskeleton architecture. The identified variant segregates with NSHL in the index Ghanaian family. The data support this nonsense variant as pathogenic, likely to impact the homeostasis of ions, solutes, and other molecules, compromising membrane barrier and signaling in the inner ear spaces.

## 1. Introduction

Hearing loss (HL) is a heterogenous sensory disorder affecting close to 5% of the world population, of which 34 million are children [1]. Systematic reports have implicated population growth (60.83%) and aging (35.35%) as key factors driving the increasing incidence [2,3,4]. However, Asia, Asia Pacific, and Africa have a higher HL burden compared to high-income countries [2]. For example, whereas there are 1–6 HL cases in every 1000 live births in Africa [5], 1.33 HL cases in 1000 births are recorded in advanced countries [4]. Auditory system defects can have profound effects on the affected individual’s quality of life; spoken language acquisition, effective communication, social and emotional challenges, among others [6]. Empirical studies have shown that early detection, accurate diagnosis, and timely intervention improve speech and language development, as well as the cognitive and psycho-social well-being of individuals affected with HL [7,8,9].

However, the diagnosis of non-syndromic hearing loss (NSHL) is mostly missed, delayed, and detected post-language acquisition age, particularly in low-to-middle-income countries. Notably, the reported mean age at medical diagnosis is 3.3 years in Cameroon [10], between 6 and 11 years in Ghana [11], 12 years in Mali [12], 3.59 ± 2.27 years in Senegal [13], and 1.25 to 3.17 years in South Africa [14]. Systematic epidemiological investigations have well established the contribution of both acquired and genetic etiology in HL in high-income countries [3,4,15]. The current advancement and progressive improvement in maternal and child healthcare services (immunization and vaccination) have apparently revealed inherent genetic factors, accounting for 50–80% congenital HL, in high-income countries [16].

Despite the reported spectrum of genetic and allelic heterogeneity in NSHL pathogenic variants across populations [4], close to 90% of cases are monogenic [4,15,17]. To date, more than 150 loci have been mapped to approximately 160 NSHL genes globally [18], mainly inherited via autosomal recessive (AR, 80%), autosomal dominant (AD, 17%), with X-linked and mitochondrial patterns described in rare situations [19]. Yet, the genetic characterization of HL is still missing in most African populations [20], highlighted by the recent studies in Ghana and Mali, that illuminated even higher degrees of genetic and allelic heterogeneity compared to advanced settings [21,22].

We present, in this report, whole-exome sequencing (WES) analysis of data obtained from a consanguineous multiplex Ghanaian family, negative for the predominant genetic cause of HL (*GJB2*-p.Arg143Trp) in Ghana. The identified bi-allelic *MARVELD2* nonsense frameshift duplication variant [c.1058dup-p.(Val354Serfs*5)] co-segregates with likely post-lingual NSHL in the family. Experimental data from cell-based immunofluorescence and immunocytochemistry studies further validate the variants’ pathogenicity and molecular underpinnings.

## 2. Results

### 2.1. Family and Phenotype Description

Of the three affected kindreds (Figure 1A,B), participants IV:3 and IV:5 were found to have post-lingual profound NSHL at the time of enrollment and sample collection. These two participants (affected females) were documented to have developed hearing disorder at age 12 years (self-reported, stopped mainstream school after 12 years). The last affected male (Figure 1A,B: IV:6) at the time of interview was diagnosed to be “hard of hearing”, with onset of the hearing disorder at age 8 years. The first two affected individuals in the pedigree (Figure 1A: IV:3 and IV:5) had profound HL, with pure-tone air and bone conduction (PTA) test average thresholds of 100 and 98 dB, respectively, as depicted in Figure 1B. Unlike the first two siblings, the third affected sibling’s (male, IV:6) hearing evaluation showed severe HL with a PTA average of 63 dB (PTA showing high-frequency HL), as seen in Figure 1B. Notably, the presented audiogram (Figure 1B) was assessed at ages 32, 28, and 26 years, respectively, for the proband (IV:3), affected sister (IV:5), and affected brother (IV:6) at the time of recruitment. All three affected siblings are good at lip reading and had effective verbal communication prior to condition onset and characterized as non-progressive NSHL. Markedly, the availability of periodic audiological reports at different ages for all affected individuals would have supported the self-reported post-lingual clinical history and the one-time audiological assessment presented. All unaffected family clinically evaluated members had normal hearing (Figure 1A, individuals III;5. III:5, IV:1, IV:2, IV6, and V:1).

### 2.2. WES Variant Identification

The WES achieved 96% of the target regions, having more than 30× coverage. WES data analysis using filtering criteria as detailed in the methods discovered the novel variant in *MARVELD2*: c.1058dup;p.(Val354Serfs*5) on exon 2 as a putative cause of the observed phenotype in homozygous state in all three affected siblings (Figure 1A,C: IV.3, IV.5, and IV.6). Though the variant (*rs2150915254*) has been reported on gnomAD with an allele frequency of 6.197 × 10^−7^ (1/1613794) (https://gnomad.broadinstitute.org/ accessed on 10 January 2025), it is absent on Bravo (https://bravo.sph.umich.edu/ accessed on 10 January 2025), and all other relevant genomic resource databases. A CADD score of 28.5 indicates pathogenic based on threshold and points established elsewhere [23,24], showing high confidence probability of loss of function (pLoF). Following the American College of Medical Genetics and Genomics (ACMG) guidelines for variants interpretation and annotation [25], the variant was classified as pathogenic (score: PSV1, PS4, PP1, PP3, PM4, and PM2), and segregates with the phenotype within the family, further supporting it as the likely causal variant associated with autosomal recessive deafness-49 (DFNB49) phenotype. This frameshift variant is predicted to be targeted by nonsense mediated decay [26], and if partially expressed, it culminates in truncation of the synthesized protein to lose a cytoplasmic domain, supporting the associated pathology.

### 2.3. Validation of WES Variant

Bi-directional Sanger sequencing analysis of the three affected and five unaffected kindred confirmed the presence of the *MARVELD2*: c.1058dup;p.(Val354Serfs*5) in a homozygous state (all affected) and heterozygote carrier and/or homozygous wild type (unaffected) individuals. The variant was absent in the 151 unrelated simplex congenital NSHL individuals aged between 5 and 26 years (mean age of 17.7 years) with equal proportion on gender (51.6% (78/151) females).

In addition, the Deafness Variation Database (https://deafnessvariationdatabase.org/gene/MARVELD2: accessed on 26 July 2024) has listed 3437 *MARVELD2* variants [27], and the CADD Phred score and classification are represented as a box and whisker plot shown in Appendix A. At the protein level, as shown in Figure 2, STRING protein—protein interaction (PPI) enrichment analysis (https://string-db.org/: accessed on 23 July 2024) produced significant interactions. PPI enrichment *p*-value: <1.0 × 10^−16^ (number of nodes = 11, number of observed edges = 52, expected number of edges = 11), which implies that these proteins at least have partial biological connection as a group than what would have been expected for a random set of proteins of the same size and degree distribution drawn from the genome. In this case, only MARVELD2 was queried, and STRING retrieved the 10 additional proteins by default, showing network and pathway interactions around the target protein. Gene ontology references and annotations converged with MARVELD2 suggested to be involved in cell population proliferation, DNA-template transcription, cytoskeleton, and protein-containing complex. This further supports the prelingual sensorineural and autosomal recessive disease–gene association in the human phenotype ontology (HPO) database. Genotype-Tissue Expression (GTEx) data (https://www.gtexportal.org/home/: accessed on 26 July 2024) validate MARVELD2 expression in 31 primary and 53 secondary tissues, underscoring the gene conserve function in humans. Appendix A details the synthesized review of *MARVELD2* NSHL reported variants worldwide.

### 2.4. Evolutionary and Conservation Analysis of MARVELD2 Protein

To evaluate the pathological effect of the frameshift variant, and the general conservation of the implicated MARVELD2 domain sequence in different species were interrogated. The retrieved protein sequence of 20 different species together with the mutant MARVELD2 sequence were aligned by ClustalW multiple sequence alignment, and the domain containing the mutation of interest analyzed (Figure 3F).

The domain impacted by this duplication and subsequent nonsense mediated decay represents a conserved region about~100 aa residues long within eukaryotic occludin protein and RNA polymerase II elongation factor ELL. Occludin is an integral membrane protein that localizes to tight junctions, while the ELL elongation factor is known to increase the catalytic function of RNA polymerase II transcription by regulating the transient pausing of the polymerase at multiple sites along the DNA strand, as well as mediating protein–protein interactions.

### 2.5. MARVELD2 Protein Modeling Analysis

We obtained a high-quality model for the wild-type MARVELD2 protein (PDB ID: 5N7K) from the PDB database (Figure 3A). Modeled novel mutant p.(Val354Serfs*5) protein based on the 5N7K template, produced a Ramachandran plot favored and outlier scores of 99.10% and 0.00%, respectively (Figure 3B). The quality estimated scores of the mutant model were within the acceptable range (Figure 3C–E). Aligning the wild-type (shown in grey) to the mutant protein (shown in red) demonstrates the quality of mutant protein model obtained (Figure 3A–E). If expressed, the variant results in a truncated and predicted non-functional MARVELD2 protein. As expected, the variant caused a truncation in the protein length as apparent in the superimposed structures (Figure 3A; superimposed). Domains search on the InterProScan server revealed that the truncation results in the loss of the cytoplasmic domain, which contains the occludin domain, necessary for localization to tight junctions [28]. This was also demonstrated by the folding model of the protein produced using Protter shown in Figure 3A, Domains (https://wlab.ethz.ch/protter/, accessed on 9 August 2024).

### 2.6. Investigating Variant MARVELD2 Truncation, Stability, and Protein-to-Protein Interactions

Ectopic expression followed by Western blot (WB) confirmed the in silico predicted truncation of the mutant protein by~100 aa (37 kDa) relative to the WT protein 588 aa~64 kDa) (Figure 4A, Appendix A), and significantly decreased expression by quantification compared to the β-actin internal control, indicated by a much less prominent band detected by WB, as well as reduced yellow fluorescence protein (YFP) fluorescence (Figure 4C, Appendix A). This demonstrates that the putative nonsense *MARVELD2* thymine duplication not only truncates protein translation but also significantly affects the stability of the mutant protein. Resolved high-resolution structure of occludin C-terminal domain shares sequence homology with MARVELD2 occludin-ELL cytosolic C-terminal domain [29]. Thus, MARVELD2 occludin-ELL cytosolic domain contains similar structural components and domain-specific residues required for zonula occludens-1 (ZO-1) complex interaction [30], crucial for membrane integrity. This domain has been well-characterized and shown to bind scaffolding protein ZO-1, anchoring it to intercellular junctions. To examine the binding ability of the mutant MARVELD2 to ZO-1, pull-down assay was performed. We found that the physical interaction between these two partner proteins remains stable despite the mutant MARVELD2 protein alteration, suggesting that this interaction may not be exclusive via the extended C-terminal domain of MARVELD2 as alluded to in the literature (Figure 4B). This observation contradicts the in silico predicted truncated domain but supports the notion that there are other redundant and/or compensatory protein domains interacting with these family of scaffolding associated proteins aside the C-terminal cytosolic domain (shared homology).

### 2.7. Cell Morphology, In Vitro Localization, and Cytoskeletal Arrangements

By similarity, the C-terminal cytoplasmic domain of MARVELD2 interacts with ZO-1, angulin-1 (LSR), angulin-2 (ILDR1), and angulin-3 (ILDR2), which recruits and anchors it to tricellular contacts. To infer probable biological significance of the mutant truncated C-terminal cytoplasmic domain, we studied and compared cellular morphology, protein localization, and cytoskeletal assembly of cells ectopically expressing the YFP-tagged WT versus the MT MARVELD2 protein. In both instances, fluorescent expression was absent from the nucleus, as can be seen in the “merged” images where no overlap of Hoechst (blue) and YPF-tagged WT/MT MARVELD2 is observed (Figure 5, Appendix A). While WT protein appears to be fairly and evenly distributed, and continuous along cellular margins, mutant expression seems sequestrated within vesicles and/or Golgi-like structures, and only sporadically dense at the cellular margin as attachments (Figure 5 and Appendix A).

However, it is important to note that higher resolution (>200 nm) than the resolution used in the present image may provide more details to clarify our assertion and the probable MARVELD2 protein and plasma membrane association. In addition, co-labeling with a Golgi-specific organelle marker not performed in this study could reveal and illuminate the unclear and/or patchy protein sequestration in the Golgi structures (Figure 5 and Appendix A, MT panel). More importantly, a substantial cytoplasmic component of the WT protein signal could not be explicitly allocated to specific subcellular compartments (Appendix A). Figure 6 and Appendix A establish counter-staining of nuclei, EYFP, ZO-1, and phalloidin (F-actin), showing cytoskeletal contact sites between adjacent and connected cells. The images reveal WT proteins localization to be less prominent near the cellular margins relative to the MT proteins that showed consistent but diffused distribution, indicated by white arrows (Figure 6 and Appendix A). The observed patches of cortical actin arcs (in phalloidin panel), pockets of stress fibers and clear positive spots at cellular contacts were consistent with exogenous MARVELD2 expression. However, at the tricellular junctions (cell vertices), WT exhibits much narrower gaps (ZO-1 and merged panel, indicated by white broken circle) compared to the wider gaps seen in cells expressing mutant protein, which suggests a plausible compromised MARVELD2, ZO-1 and occludin-1 interactions impacting complexes formation and integrity of cellular junctions (Appendix A).

## 3. Discussion

This study performed an in-depth genetic and molecular characterization of a bi-allelic *MARVELD2*: c.1058dup;p.(Val354Serfs*5) variant identified through WES of a multiplex consanguineous family segregating severe-to-profound [22]. The report expands the number of *MARVELD2* variants associated with NSHL worldwide to 13, and contributes to the global disease–gene pair curation [30,31,32,33,34,35,36,37,38,39].

The varied hearing threshold observed in the index family replicates some described cases in other populations [30,35]. However, factors linked to the reported inter-and intra-kindreds phenotype variation and penetrance remains unclear [30,32,39]. While some authors attribute the phenotype variability to environmental factors [35], the likely contribution of inherent genetic modifiers, modulating the varied phenotype thresholds among affected kindreds remains unexplored [30]. We report a non-progressive phenotype with no correlation with age [38], supporting similar observations in prior studies [30,32]. The subtle differences in age of onset of *MARVELD2* variants associated with the ARNSHL phenotype may reflect the diversity and population-specific variants, as well as the distinct pathogenic mechanisms. Aside the most common and re-occurring *MARVELD2*:c.1331+2T>C variant that has been reported in nine (9) different families in different regions [30,31,32], all other pathogenic *MARVELD2* variants reported to date are population-specific, further supporting the differences in clinical manifestation.

Markedly, the cellular characterization evaluated the credible biological consequence of the identified variant predicted to truncate a conserved cytoplasmic domain, and the probable impact on protein structure and function. First, the *MARVELD2*: c.1058dup;p.(Val354Serfs*5) variant causes a frameshift and is predicted to be targeted by nonsense mediated decay (NMD), with the resulting premature termination codon (PTC) located more than 50–55 nucleotides upstream of the final exon–exon junction, which fulfil the criteria for NMD activation described elsewhere [26]. Next, if the mutant protein is still expressed at some level, this frameshift variant found truncates the occludin_ELL C-terminal cytosolic domain of MARVELD2, involved in signaling relay and interactions with a family of scaffolding membrane-associated proteins, including ZO-1, ZO-2, and ZO-3 vital in cell barrier functions [29,40].

Similar to the identified variant, exon 2 pathogenic variants that delete and/or truncate the complete translation of the 588 amino acids syntheses have been linked to MARVELD2 protein LoF [41]. Predominantly, of the six *MARVELD2* coding exons, exon 2 encodes four out of the six transmembrane domains in the protein [40], lending further support to the pathogenicity of exon 2 deleterious variants like the present report. MARVELD2 expression regulates the size and selectivity of the tricellular central tube pore, which allows for the passage of different molecules and macromolecules [42]. Particularly, in the inner ear compartments, the transport of sodium and potassium are tightly balanced between the perilymph and endolymph.

In vivo, the variant reported in this study together with four (4) exon four splice site [30], and two nonsense [32,41] *MARVELD2* variants, are predicted to culminate in no protein expression due to NMD, and even if expressed, the truncated protein is likely to be at very minimal levels and/or LoF [26]. The underlying pathobiology is in tandem with most reported causal variants predicted to have a pathogenic effect in other populations; ranging from nonsense protein truncation [32,41] to in-frame deletions [35] and splice site variants that alter mRNA processing and translation [43]. Most *MARVELD2* pathogenic variants that affect the transmembrane domain are chiefly in the cytoplasmic tail [38,43,44]. Akin to the predicted truncated protein caused by the *MARVELD2*: c.1058dup;p.(Val354Serfs*5) variant, the MARVELD2, if translated, will lack a portion of the fourth transmembrane domain and cytosolic tail, analogous with other pathogenic variants described in Pakistan [30,35,43], Iran [37,38,41], Czech Roma [31], and Slovak Roma [34].

We produced cellular constructs carrying the variant of interest to investigate MARVELD2 truncated protein expression levels to infer molecular function. The observed differential expression evident in the Western blot reflects the predicted variant effect (nonsense mediated decay of the mutant mRNA), strengthening the in silico predicted mutant protein truncation, likely to affect protein levels and function. Interestingly, the predicted C-terminal cytosolic domain loss (~100 amino acids), if translated, showed no observable differences between the targeted domain binding and interactions (between the mutant and wild-type proteins) from the co-immunoprecipitation assay. This observation adds support for the suggested NMD cellular quality control activation and mechanisms, likely to degrade the mutant mRNA to minimize aberrant protein synthesis [26].

Mice homozygous for the knock-in pathogenic *Marveld2* variant exhibited syndromic deafness with rapid progressive degeneration of the hair cells, increased body and organ weights, and abnormal tTJs with vestibular disturbance [45]. Riazuddin and colleagues demonstrated HL association with *MARVELD2* mutations in humans and mice, and generated *Marveld2^−/−^* mice that displayed early onset progressive HL [30]. In a comparable phenotype expression study that used a *Marveld2* knock-in, the investigators observed additional pathologies in varied organs other than the inner ear [35]. Cochlea hair cells damage and epithelial barrier function disturbance in the inner ear was cited as the likely molecular basis for the phenotype development [40]. The spectrum of hearing disorder manifestation in modeled mice conflicts with human phenotypes, where no other organ involvement has been reported [30,31,32]. The complex expression of redundant TJs in humans, understood to evade the wide phenotype characteristic of the *Malverd2* mutant mice, has been cited as the basis of observed inconsistencies in *Marveld2*-deficient mice and DFNB49 human phenotypes [35]. Prior *Marveld2*^−/−^ mice studies have demonstrated early onset deafness with multiple-organs disorder and vestibular involvement. However, neither the multi-organ nor vestibular dysfunction was observed in the hearing-impaired individuals assessed in the index family.

## 4. Materials and Methods

### 4.1. Ethical Approvals

This study was conducted following the ethical principles of the declaration of Helsinki guiding medical research. This study received approval from the Noguchi Memorial Institute for Medical Research Institutional Review Board (NMIMR-IRB CPN 006/16-17), the Faculty of Health Sciences’ Human Research Ethics Committee, University of Cape Town (HREC 104/2018), the Columbia University’s Institutional review board (IRB-AAAS2343), and the University of Ghana’s Ethics Committee for Basic and Applied Sciences (ECBAS 053/19-20). Prior to enrollment of the participants, the protocol was explained in the participants’ preferred language and sign language interpretation used for affected individuals for full comprehension, signed and verbal consent obtained before recruitment. Parental consent was obtained for one individual aged under 18 years.

### 4.2. Participants’ Enrollment and Sample Collection

The participants enrolled in this study are members of a large Ghanaian family segregating apparently a post-lingual NSHL. The proband and two affected siblings in this multiplex consanguineous family were recruited through our community engagement and recruitment program. Medical records and clinical history of the affected siblings were thoroughly examined by a clinical geneticist, and a specialist; ear, nose, and throat (ENT). Participants’ clinical history and responses to a structured questionnaire, which was developed for this study, were used to assess proband and affected siblings of suspected known environmental factors that may contribute to the phenotype. All three affected and five unaffected kindred members underwent audiometry pure-tone, air and bone conduction and their hearing threshold average evaluated for both ears.

### 4.3. Whole Exome Sequencing

The affected individuals previously tested negative for the gap junction protein beta 2 (*GJB2*) founder variant (*GJB2*:c.427C>T-p.Arg143Trp) predominant in Ghana, and had no pathogenic variant identified in the *GJB2* coding region [11]. Three affected and one unaffected kindred gDNA underwent WES at Omega Bioservices, Inc., Norcross, GA 30093-2243, USA) using pair-end 150 bp run format on an Illumina HiSeq 2500 platform. Sequence libraries were prepared using Nextera Rapid Capture Exome kit^®^ (Illumina, San Diego, CA, USA), after which they were hybridized with a 37 Mb probe pool to enrich exome sequences as described elsewhere [46]. Generated libraries of sizes between 300 bp and 421 bp were quantified using quantitative PCR (qPCR) before sequencing.

### 4.4. Annotation and Filtering Strategy

The obtained reads were analyzed following the Illumina DRAGEN Germline Pipeline v3.2.8. After an in-house quality control check, the resultant high-quality reads were aligned to the genome reference consortium human built 37 (GRCh37/hg19) using the Illumina DRAGEN 05.021.408.3.4.12 software. Sorting and marking of duplicates were performed using Picard. The Genome Analysis Toolkit (GATKv4.1.7) software package [47] was used to conduct joint variant calling for single-nucleotide variations (SNV) and Insertion/Deletions (Indels). To solve the possible genetic marker co-segregating with the late-onset ARNSHI in this multiplex consanguineous family, the genome analysis toolkit (GATK) software v4.0.6.0 was used to performed joint variants calling. And the identified variants were annotated and filtered using ANNOVAR version 2021Oct19, as described by Wang et al. [47], following analysis strategy described previously [22,25]. Filtering of SNVs and indels was performed using the Genome Aggregation Database (gnomAD) after checking known pathogenic variants for NSHI regardless of their frequency [20], with a population-specific minor allele frequency of <0.005 [for homozygous and potentially compound heterozygous variants and variants on the X chromosome AR] and <0.0005 for heterozygous variants. Synonymous and intronic variants that were not close to a splice site region were removed. Variants that met the above criteria were further prioritized based on in silico prediction scores from Sorting Intolerant From Tolerant (SIFT); MutationTaster; combined annotation dependent depletion (CADD); Genomic Evolutionary Rate Profiling (GERP++); polymorphism phenotyping v2 (PolyPhen-2); and deleterious annotation of genetic variants using neural networks (DANN). The variants were further assessed with information from the Hereditary Hearing Loss Homepage (HHL), Online Mendelian Inheritance in Man (OMIM), Human Phenotype Ontology (HPO), ClinVar, and allele frequencies assessed using the TOPMed Bravo database.

The autosomal recessive mode of inheritance was first considered for the variant filtering since it conformed to the pedigree structure. For the identified putative variant, we considered the following: (1) occurred in known HI genes; (2) had a predicted effect on protein function or pre-mRNA splicing (nonsense, missense, start-loss, frameshift, splicing, start-loss); and (3) co-segregated with the HI phenotype within the family. The American College of Medical Genetics and Genomics and Association of Molecular Pathology (ACMG-AMP) guidelines for HL [48] were followed to evaluate clinical significance. Copy number variants (CNVs) were annotated and filtered with AnnotSV [49] and an in-house pipeline that interrogates BioMart [50] and the Database of Genomic Variants [51].

### 4.5. Sanger Sequencing

Sequence specific primers for the *MARVELD2* were designed using the National Center for Biotechnology Information (NCBI) primer Basic Local Alignment Search Tool (BLAST), covering the region with our variant of interest. Designed primers were optimized using the Oligo-analyzer tool designed by Integrated DNA Technologies, Inc. (Coralville, IA 52241, USA). *MAERVELD2* exon 2 was PCR amplified using designed primer pair, forward: (5′-GAGAATACTGGGTGTGGTGGAG-3′) and reverse: (5′-ACAGACATAATAATGCAGCCAACAA-3′), using the cycling conditions denaturation, annealing, and extension (95 °C, 60 °C, and 72 °C), respectively. Bi-directional Sanger sequencing with the BigDye^TM^ Terminator v3.1 Cycle Sequencing Kit was performed on PCR amplicons and sequence analyzed using ABI 3130XL Genetic Analyzer^®^ (Applied Biosystems, Foster City, CA, USA). FinchTV v1.4.0 and UGENE v34.0 [52] were used to check Sanger sequenced ABI files quality and analyzed as described by Okonechnikov and colleagues [52]. The implicated *MARVELD2*: NM_001038603.3:c.1058dup;p.(Val354Serfs*5) variant was investigated in 151 unrelated simplex Ghanaian probands with NSHL.

### 4.6. Evolutionary Conservation

To further interrogate the MARVELD2 substituted amino acid and domain conservation in different species, evolutionary conservation of the affected domain was assessed using phyloP, SiPhy, and Genomic Evolutionary Rate Profiling (GERP), and phastCons scores. Using the UGENE v34.0 software [52], ClustalW multiple alignment was performed with the downloaded protein sequences and *MARVELD2*:c.1058dup;p.(Val354Serfs*5) variant protein sequence.

### 4.7. MARVELD2 Protein Model Analysis

The wildtype protein model solved by Schuetz et al. 2017 [53] was retrieved from the protein database (PDB) and visualized in PyMOL version 2.5. A homology protein modeling approach was employed to model the mutant protein using a web-based tool, SWISS-MODEL [54]. The wildtype protein (PDB ID: 5N7K) was used as a template for mutant protein modeling. Solved mutant predicted structure was downloaded from the SWISS-MODEL and visualized in PyMOL, and the protein alignment function of PyMOL used to align the wild-type protein to mutant protein.

### 4.8. HEK-293 Cell In Vitro Functional Assay of the MARVELD2 Variant

#### 4.8.1. Expression Vectors and Site-Directed Mutagenesis

Mammalian expression plasmid construct containing full length human tricellulin/*MARVELD2*; huTriC-pCMV10-DDK-FLAG and huTriC-N-EYFP-pcDNA-3 were generously gifted by Prof. Sussane Krug (Institute of Clinical Physiology, University of Berlin, Berlin, Germany). Whereas the pCMV10 plasmid had DDK-FLAG expression tag, the pcDNA-3 had a YFP N-terminal tag, with both plasmids’ harboring ampicillin and neomycin selectable markers (Appendix A). The cloning of huTriC-pCMV10-DDK-FLAG and huTriC-N-EYFP-pcDNA-3 has been described elsewhere [42]. Recombinant plasmids produced were extracted with Pure Yield^TM^ Plasmid Miniprep System (Promega Cooperation, Madison, WI, USA) and the full-length human tricellulin (*MARVEDL2*) coding sequence insert was validated by Sanger sequencing (Inqaba Biotech, Johannesburg, South Africa).

To create our variant of interest, site-directed mutagenesis (SDM) sequence-specific primer pairs (forward primer: 5′-CCTGTTTGTCACCATGATTAGTTTATCTCATTAGTGC-3′ and reverse primer: 5′-GCACTAATGAGATAAACTAATCATGGTGACAAACAGG-3′) were designed for both plasmids targeting the human full-length sequence using the QuickChange Site-Directed Mutagenesis Kits following suppliers’ guidelines with minor modifications. The *MARVELD2* mutant constructs for both plasmids were created using the Invitrogen^TM^ Platinum^TM^ SuperFi^TM^ II PCR Master Mix (Invitrogen, ThermoFisher Scientific, Washington, DC 98103, USA), with the conditions denaturation, annealing, and extension at 98 °C, 68 °C; and 72 °C for 25 reaction cycles. Purified recombinant mutant *MARVELD2*; huTriC-pCMV10-DDK-FLAG and huTriC-N-EYFP-pcDNA-3 products were cloned in XL1-Blue and DH5-α *E. coli* cells, respectively, and *MARVELD2*:c.1058dup-pVal354Ser*5 duplication variant confirmed in both constructs (Appendix A) by Sanger sequencing (Inqaba Biotec, Gauteng, South Africa).

#### 4.8.2. Cell Cultures and Transfections

Plasmid constructs (wild-type and mutant) were transfected into immortalized human embryonic kidney cells (HEK-293 cells) cultured in Dulbecco’s Modified Eagle Medium (DMEM) (ThermoFisher Scientific, Waltham, MA 02451, USA) supplemented with 10% (*v*/*v*) fetal bovine serum (FBS) (ThermoFisher Scientific, Waltham, MA 02451, USA) and 1% (*w*/*v*) penicillin/streptomycin (Sigma-Aldrich, Missouri, USA), in a humidified incubator at 37 °C and supplemented with 5% carbon dioxide (CO_2_). Plasmid transfections were performed using the X-tremeGene^TM^ HP DNA Transfection Reagent (Sigma-Aldrich, Missouri, MO 63178, USA) following the manufacturer’s instructions. Briefly, 1.0 µg of respective plasmid DNA was mixed with 3 µL of transfection reagent according to the manufacturer’s protocol (huTriC-pCMV10-DDK-FLAG: Wild-Type or Mutant; and huTriC-N-EYFP-pcDNA-3: Wild-Type or Mutant) and transfected into HEK-293 cells plated ±18 prior at a density of 2.5 × 10^5^ cells/mL, Appendix A. Total protein was isolated 48 h post-transfection for downstream applications including Western blotting and co-immunoprecipitation (Co-IP).

#### 4.8.3. Western Blot Analysis

HEK-293 cells were transfected as described above with 1.0 µg plasmid DNA (huTriC-pCMV10-3xFLAG: wild-type or mutant). Forty-eight hours post-transfection, cells were lysed in (150 nM NaCl, 1% Triton X-100, 0.1% SDS, 10 nM Tris pH 7.5 and 1% Deoxycholate powder) combined with 7X complete™, Mini, EDTA-free, Protease Inhibitor (Roche), and protein concentration determined using BCA (Pierce™ BCA Protein Assay). An amount of 100 µg of protein was used for pulldown for both mutant and wild-type extracts using antibodies against ZO-1 (4 µg), FLAG (2 µg) (tagged to MARVELD2) and IgG (4 µg) as negative isotype control. Primary antibody incubation was performed at 4 °C overnight with gentle agitation, followed by incubation with 50 µL of protein A agarose beads slurry (Roche, Basel, Switzerland) for 4 h. Beads were collected by centrifugation, washed, and resuspended in 50 µL 2× Laemmli buffer. Bound proteins were detached from beads by denaturation at 95 °C for 5 min prior to SD-PAGE separation and Western blot analysis.

#### 4.8.4. Co-Immunoprecipitation Assay

HEK-293 cells were transfected as described above with 1.0 µg plasmid DNA (huTriC-pCMV10-3xFLAG: wild type or mutant). Forty-eight hours post-transfection, cells were lysed in (150 nM NaCl, 1% Triton X-100, 0.1% SDS, 10 nM Tris pH 7.5 and 1% Deoxycholate powder) combined with 7X complete™, Mini, EDTA-free, Protease Inhibitor (Roche, Basel, Switzerland), and protein concentration determined using BCA (Pierce™ BCA Protein Assay). An amount of 100 µg of protein was used for pulldown for both mutant and wild type extracts using antibodies against ZO-1 (4 µg), FLAG (2 µg) (tagged to MARVELD2) and IgG (4 µg) as negative isotype control. Primary antibody incubation was performed at 4 °C overnight with gentle agitation, followed by incubation with 50 µL of protein A agarose beads slurry (Roche, Basel, Switzerland) for 4 h. Beads were collected by centrifugation, washed, and resuspended in 50 µL 2× Laemmli buffer. Bound proteins were detached from beads by denaturation at 95 °C for 5 min prior to SD-PAGE separation and Western blot analysis.

#### 4.8.5. Fluorescent Microscopy

HEK-293 cells were seeded in a glass bottom 35 mm culture dish and transfected as described above with wild-type or mutant huTriC-N-EYFP-pcDNA-3 plasmids. Confocal microscopy imaging was performed at 48 h post-transfection using a confocal laser scanning microscope, Carl Zeiss LSM880 Airyscan, operated by ZEN Black software v2.3, using a 63×, numerical aperture 1.4 objective and images captured analyzed using the ZEISS ZEN Lite Image Acquisition and Analysis Software Version 3.10 (Carl-Zeiss-Promenade 10, Carl Zeiss, Oberkochen, Germany).

#### 4.8.6. Immunocytochemistry

HEK-293 cells were seeded on sterilized poly-L-lysine-coated coverslips and transfected as described above with the appropriate plasmids. At 48 h post-transfection, cells were fixed in 2% paraformaldehyde (PFA) for 15 min, permeabilized with 0.2% Triton X-100, and blocked for 1 h in blocking buffer (1% BSA in PBS), followed by 4 °C overnight incubation in primary antibody diluted in blocking buffer as follows: mouse monoclonal ZO-1 antibody, 1:200 (33-9100, Invitrogen, Thermo Fisher Scientific), Oregon Green ^TM^ 488 Phalloidin, 1: 1000 (Invitrogen, Thermo Fisher Scientific). Cells were thereafter incubated in secondary antibodies for 1 h at RT as follows: goat anti-mouse IgG H&, Texas Red secondary antibody, 1:1000 (Abcam, Cambridge, United Kingdom). Nuclei were counter stained with Hoechst (4-[6-[6-(4-methylpiperazin-1-yl)-1H-benzimidazol-2-yl]-1H-benzimidazol-2-yl] phenol; trihydrochloride) and coverslips mount in Mowiol containing 2.5% 1,4-diazobicyclo-[2.2.2]-octane (DABCO, D2522, Sigma-Aldrich, Missouri, MO 63178, USA). Slides were viewed using a confocal laser scanning microscope (LSM 510 Meta, Carl Zeiss, Jena, Germany) using 543 and 488 nm excitation wavelengths and captured images viewed and analyzed using ZEISS ZEN Lite Image Acquisition and Analysis Software Version 3.10 (Carl-Zeiss-Promenade 10, Carl Zeiss, Oberkochen, Germany).

## 5. Conclusions

We report a bi-allelic pathogenic *MARVELD2*:c.1058dup;p.(Val354Serfs*5) variant associated with NSHL in a multiplex consanguineous family from Ghana. NMD activation likely degrades translationally abnormal mutant MARVELD2 mRNAs and/or if expressed may be at minimal levels, as well as the loss of conserved cytosolic C-terminal domain. This can impair MARVELD2 normal localization to bTJs and tTJs via disruption of ZO-1 and other membrane-associated proteins interaction, mediating actin cytoskeleton complexes and architecture. However, the identification of the variant in an unrelated case(s) and/or population(s) will further validate this genotype–phenotype association. This study emphasized the need to explore HL genetic characterization using WES in understudied but genetically diverse African populations.

## Figures and Tables

**Figure 1 ijms-26-03337-f001:**
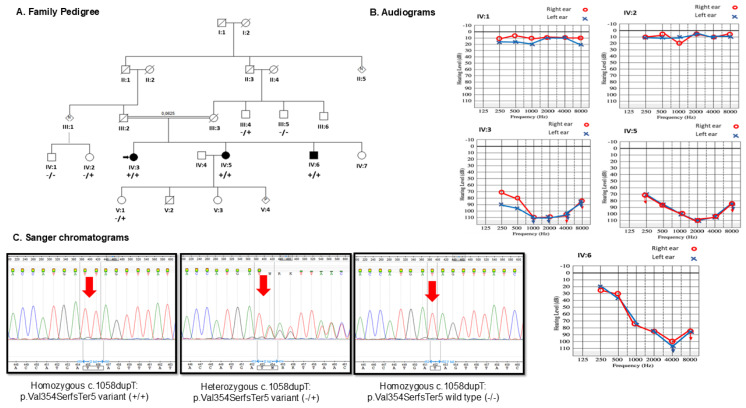
Elucidated novel bi-allelic *MARVELD2*: c.1058dupT-p.(Val354SerfsTerfs*5) variant Sanger sequencing validation, phenotype segregation and genotype–phenotype correlations. Panel (**A**). Family pedigree showing evidence of consanguinity and affected kindreds. The genotypes of the family members are represented with +/+; −/+; and −/− which denotes homozygote for the putative pathogenic variant, heterozygote, and homozygote wild type, respectively. Individuals homozygous for this mutation (IV:3, 34 years, IV:5, 31 years, and IV:6, 27 years) showed post-lingual hearing impairment, and homozygote wildtype and heterozygous siblings with normal hearing. Symbols filled denote affected persons, squares and circles represent males and females; Panel (**B**). Pure-tone Air and Bone Conduction (PTA) audiograms of the family members as indicated with the pedigree number on top of each audiogram. Carriers or family member (s) heterozygote of the variant as illustrated (IV;1 and III;5) showed normal hearing and affected kindred homozygous of the variant having severe (IV:6) to profound (IV:5) bilateral sensorineural ARNSHL; Panel (**C**). Representative Sanger sequence chromatograms of the various genotypes of *MARVELD2*: c.1058dup;p.(Val354Serfs*5) pathogenic variant. Post-lingual ARNSHL in the family co-segregate with the identified bi-allelic c.1058dupT-p.(Val354SerfsTersf*5) variant. dupT = Thymine duplication; black arrow shows proband; red arrow indicates variant location; dB = Decibels; Hz = Hertz.

**Figure 2 ijms-26-03337-f002:**
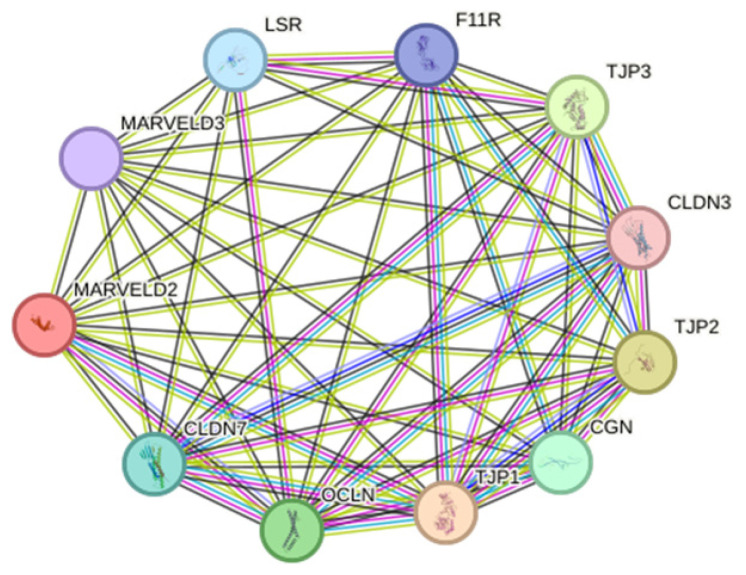
MARVEDL2 STRING protein–protein interaction (PPI) enrichment network analysis. Proteins are represented by network nodes; red colored node indicates query proteins and first shell of interactors. Filled and empty nodes denote proteins of known and unknown 3D structures, respectively. Protein–protein associations are represented as edges, and associations are near specific and meaningful (i.e., contribute or share common biological function). Known, predicted, and other interactions are also shown as associations between the set of proteins. Cellular component localization retrieved from the gene ontology network supports tricellular tight junction as the top hit, followed by apicolateral plasma membrane, bicellular tight junction, basolateral plasma membrane, and plasma membrane.

**Figure 3 ijms-26-03337-f003:**
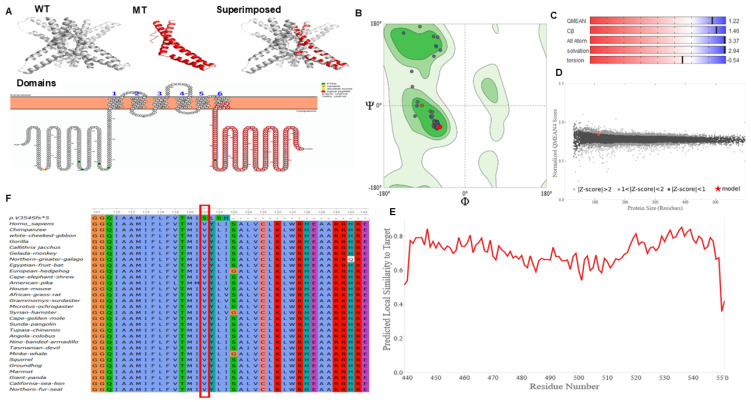
MARVELD2: p.Val354SerfsTer5 protein and wild-type MARVELD2 protein modeled structure. Panel (**A**). Structure of MARVELD2 protein: wild-type (WT), mutant (MT), superimposed (alignment of wildtype and mutant MARVELD2: p.Val354SerfsTersf*5 proteins), and Domain analysis demonstrating the truncation of~100 conserved residues (occludin domain and RNA polymerase II elongation factor (ELL)). The schematic diagram of the primary structure of the MARVELD2 protein showing the truncated cytoplasmic region is highlighted in red. Panel (**B**). Quality assessment of modeled mutant protein Ramachandran plot. Panel (**C**). Estimated modeled protein quality. Panel (**D**). Comparison of the solved mutant model with non-redundant set of PDB protein structures. Panel (**E**). Local quality estimate of the p.Val354Serfs*5 mutant protein. Panel (**F**). MARVELD2 protein sequence and amino acid schema representing multiple sequence alignment of MARVELD2 protein of different species, using ClustalW algorithm. A red rectangle is used to highlight the position of the MARVELD2: c.1058dup;p.(Val354Serfs*5) variant.

**Figure 4 ijms-26-03337-f004:**
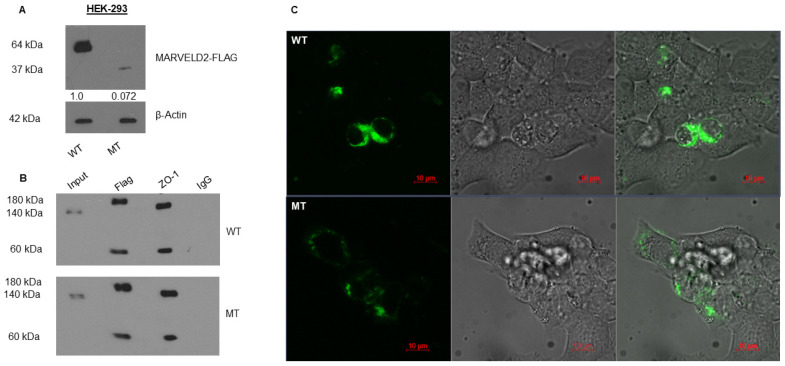
MARVELD2 wild-type and mutant protein expression and binding to ZO-1. Panel (**A**). Western blot of wild type (WT) and mutant (MT) MARVELD2 full-length protein expression levels. Western blot of total protein extracted from HEK-293 cells transfected with WT and MT: MARVELD2-pcMV10-N-FLAG plasmid constructs, probed with anti-FLAG. β-actin served as a loading control. Panel (**B**). Co-IP investigating the binding of MARVELDE2 to ZO-1 shows no disruption of interaction in mutant MARVELD2 protein. Pulled-down proteins were resolved by SDS-PAGE and stained with ZO-1 and IgG used as non-specific control. Panel (**C**). YFP fluorescence (expression) in HEK-293 transfected cells (MT or WT) 24 h post-transfection. Uncropped Western blot images added as Appendix A. Co-IP = co-immunoprecipitation; SDS-PAGE = Sodium dodecyl sulphate–polyacrylamide gel electrophoresis; and IgG = Immunoglobin G.

**Figure 5 ijms-26-03337-f005:**
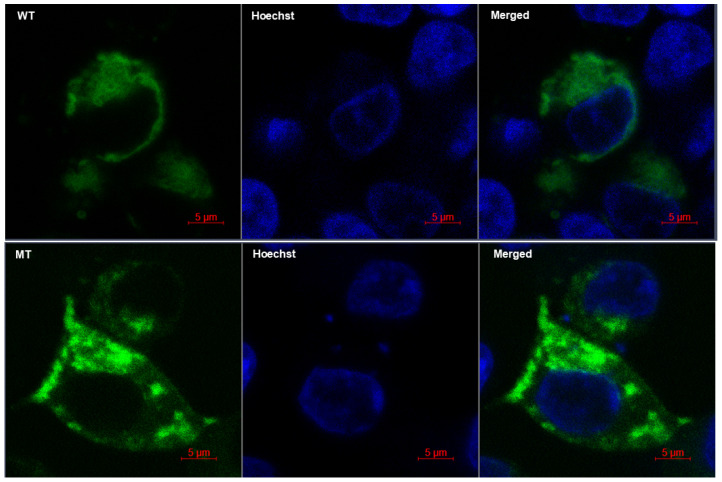
Immunofluorescence showing differentiated MARVELD2-N-EYFP tagged protein cytoplasmic and membrane-bound localization in wild-type (WT) and mutant (MT) transfected HEK–293 cells 48 h post-transfection by confocal microscopy analysis. Nuclei were stained with Hoechst (blue). Wild-type Panel: Wild-type EYFP localization is prominent in cytoplasmic inclusions, and more membrane-bound with a uniform distribution along cellular margin (EYFP green localization). Mutant Panel: Mutant expression appears patchy and sequestered. WT = Wild-type; MT = Mutant; EYFP = Exogenous Yellow Fluorescence Protein. Bars = 5 µm.

**Figure 6 ijms-26-03337-f006:**
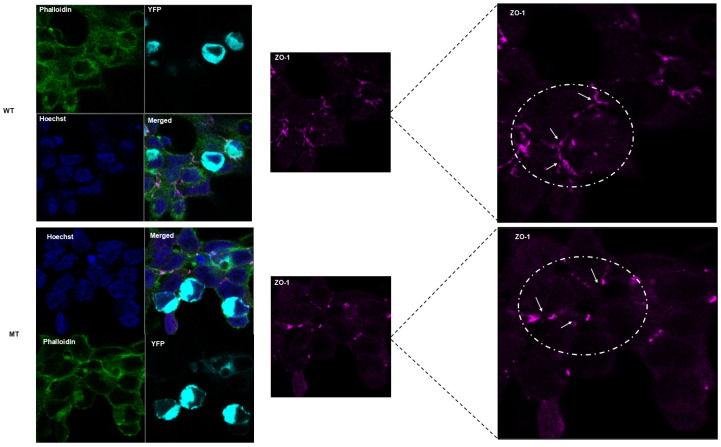
Wild-type or mutant MARVELD2-N-EYFP, zonula occluden-1 (ZO-1), and phalloidin staining along cellular margins and tricellular junctions. HEK-293 cells 48 h post-transfection localization of MARVELD2-p.Val354SerfsTer5 protein (WT or MT) were stained for ZO-1 and phalloidin, and viewed using confocal microscopy. MARVELD2 proteins were viewed using YFP channel, ZO-1 by Texas-red (depicted as violet), phalloidin actin stain using Oregon green (green), and Hoechst stain for DNA in the nuclei (blue). Wild-type panel: Wild-type transfected HEK-293 cells; localized expressed MARVEDL2-N-EYFP is condensed with consistent cell rounding, inferring intact phalloidin–actin membrane (white arrow in EYFP panel). Phalloidin-stained actin and complexes appear much predominant along stress fibers and cortical arcs (white arrows in ZO-1 panel) which is consistent with exogeneous MARVELD2 in vitro expression. Actin accumulation demonstrated by observed positive-spots at tricellular vertices, formed narrow gaps of phalloidin stain (white arrow and circle), depicting ZO-1 and MARVELD2 complexes. Mutant Panel: In mutant transfected HEK-293 cells, the localization of expressed tagged MARVELD2-p.Val354SerfsTerfs*5 seems to be diffused (EYFP panel), likely compromising MARVELD2 and ZO-1 interactions as well as its complex with other membrane-associated proteins. The phalloidin–actin membrane barrier reveals an apparent much wider gap along tricellular vertices (white circle and arrows), showing relatively less distribution of stress fibers (white arrows) and positive spots along cellular margins (white arrows) compared to the wild-type. Bars = 10 µm.

## Data Availability

All data generated or analyzed during this study are included in this published article [and its Appendix A], and for ethical concerns the individual-exome sequence data cannot be made publicly available but can be obtained upon reasonable request from the corresponding author (A.W.). The SNV data are available in the dbSNP repository (https://www.ncbi.nlm.nih.gov/snp/, accessed on 19 November 2024), and variant sequence data were deposited to ClinVar (Accession number SCV005848187).

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
