# Peer review of "Bi-Allelic MARVELD2 Variant Identified with Exome Sequencing in a Consanguineous Multiplex Ghanaian Family Segregating Non-Syndromic Hearing Loss"

_ijms, 2025, doi:10.3390/ijms26073337_

Round 1

Reviewer 1 Report

Comments and Suggestions for Authors

The study reported a novel c.1058dup (p.Val354Serfs*5) mutation in MARVELD2 from a consanguineous Ghanaian family with deafness. As concluded by the authors, the identified mutation seems to be definitely pathogenic by the evidences of genetic data, in silico prediction, and in vitro analysis. This finding will be useful for exact genetic diagnosis of nonsyndromic deafness, but its significance cannot be overemphasized because the causative mutation was identified in only one family.

OMIM registered autosomal recessive nonsyndromic deafness caused by MARVELD2 mutations to “autosomal recessive deafness-49 (DFNB49)”. Thus, it is recommended to refer DFNB49 as the clinical phenotype of the examined patients.

The authors used a term “hearing impairment” through the manuscript including title. I would like to recommend to change it to “deafness” or “hearing loss” instead of “hearing impairment”. An abbreviation “NSHI” was also non-recommendable.

In Figure 2, (B), (C), and (D) are deemed unnecessary. Please delete figures (B-D), captions, and related descriptions in the text.

In Figure 3, (E), (F), and (G) are deemed unnecessary. Please delete figures (E-G), captions, and related descriptions in the text.

Table 1 is deemed unnecessary. It is recommended to delete it and related descriptions in the text.

In Figure 4B, the molecular sizes are the same between WT and MT. Is it OK?

[Title and whole text] It is recommended to indicate gene name of MARVELD2 into italic.

[Introduction] For NSHI and WES, provide full words first before abbreviations.

What mean “multiplex” for consanguineous multiplex Ghanaian family? This word seems to be not necessary.

[Page 1, line 31-32]  Please delete following sentence: The three hearing impaired siblings were offsprings of Ghanaian parents who were first cousins.

[Page 1, line 40-41]  Identified MARVELD2:c.1058dup;p.(Val354Serfs*5) pathogenic variant > Identified variant.

[Page 4, line 144-145]  Please delete following sentence: All unaffected family members (Figure 1A) tested for the variant were either heterozygote carriers (three) and/or homozygous wild type (two) allele.

[Page 7, line 226]  Domains?

[Page 8, line 260-261] For Sodium dodecyl sulphate-polyacrylamide, insert “gel electrophoresis. And, delete “kDa = kilo Dalton; µm = Micrometer. Bars = 10 µm”.

[Page 10, line 319-330] It is recommended to delete whole section 2.8.

[Page 12, line 420-422]  Please delete following sentence: The study was performed with strict adherence to the guidelines and principles ensuring participants’ safety.

Comments on the Quality of English Language

Many redundant expressions and unnecessary descriptions are found. It is recommended to edit carefully through entire manuscript.

Author Response

  1. The study reported a novel c.1058dup (p.Val354Serfs*5) mutation in MARVELD2 from a consanguineous Ghanaian family with deafness. As concluded by the authors, the identified mutation seems to be definitely pathogenic by the evidence of genetic data, in silico prediction, and in vitro analysis. This finding will be useful for exact genetic diagnosis of nonsyndromic deafness, but its significance cannot be overemphasized because the causative mutation was identified in only one family.

Response: Authors have added the statement to the conclusion - “However, the identification of the variant in an unrelated case(s) and/or population(s) will further validate this genotype-phenotype association”. Lines 588 - 589.

  1. OMIM registered autosomal recessive nonsyndromic deafness caused by MARVELD2 mutations to “autosomal recessive deafness-49 (DFNB49)”. Thus, it is recommended to refer DFNB49 as the clinical phenotype of the examined patients.

Response: DFNB49 phenotype has been added. Lines 135-136.

  1. The authors used a term “hearing impairment” through the manuscript including title. I would like to recommend changing it to “deafness” or “hearing loss” instead of “hearing impairment”. An abbreviation “NSHI” was also non-recommendable.

Response: Thanks for this point; authors acknowledge the fact that the Deaf community prefers the term hearing loss or deafness to “Hearing impairment.” Understandably, hearing impaired is perceived as implying some deficit relative to hearing loss or deafness that reflects a unique culture and/or identity. Accordingly, “hearing impairment” has been changed to “hearing loss” in the whole manuscript.

  1. In Figure 2, (B), (C), and (D) are deemed unnecessary. Please delete figures (B-D), captions, and related descriptions in the text.

Response: Figure 2 is revised. Lines 152 – 161.

  1. In Figure 3, (E), (F), and (G) are deemed unnecessary. Please delete figures (E-G), captions, and related descriptions in the text.

Response: Figure 3 is revised accordingly. Lines 183– 192.

  1. Table 1 is deemed unnecessary. It is recommended to delete it and related descriptions in the text.

Response: Table 1 has been moved from the main manuscript to supplementary materials (Table S4).

  1. In Figure 4B, the molecular sizes are the same between WT and MT. Is it OK?

Response: Yes, it is okay. WT and MT bands are the same, indicating that the physical interaction between these two partner proteins (ZO-1 and MARVELD2) investigated remains stable despite the mutant MARVELD2 protein alteration, suggesting that this interaction may not be exclusive via the extended C-terminal domain of MARVELD2 as alluded to in literature. This observation raises further questions on previously reported MARVELD2 redundancy assertion, which requires future in vivo studies using mice model—lines 231 – 238 detail the explanation of this observation.

  1. [Title and whole text] It is recommended to indicate the gene name of MARVELD2 in italics.

Response: Changed to italics. Line 2. And italics when referring to the gene in the whole manuscript.

  1. [Introduction] For NSHI and WES, provide full words first before abbreviations.

Response: NSHL and WES, written in full on first use. Line 58 and 76.

  1. What mean “multiplex” for consanguineous multiplex Ghanaian family? This word seems to be not necessary.

Response: Multiplex, as used in the manuscript, refers to a description of the number of individuals affected in the investigated family (more than one affected individual, denoted as multiplex), and “simplex’, indicates an isolated or only one person is affected in the family investigated.

  1. [Page 1, line 31-32] Please delete following sentence: The three hearing impaired siblings were offsprings of Ghanaian parents who were first cousins.

Response: Sentence deleted as suggested (Abstract lines 31 – 32).

  1. [Page 1, line 40-41] Identified MARVELD2:c.1058dup;p.(Val354Serfs*5) pathogenic variant > Identified variant.

Response: The sentence is rephrased accordingly and now reads: “The identified variant segregates…” Line 39 – 40.

  1. [Page 4, line 144-145] Please delete following sentence: All unaffected family members (Figure 1A) tested for the variant were either heterozygote carriers (three) and/or homozygous wild type (two) allele.

Response: the sentence is removed. Line 142 – 143.

  1. [Page 7, line 226] Domains?

Response: Domains. Line 211.

  1. [Page 8, line 260-261] For Sodium dodecyl sulphate-polyacrylamide, insert “gel electrophoresis. And, delete “kDa = kilo Dalton; µm = Micrometer. Bars = 10 µm”.

Response: “gel electrophoresis” inserted, and kDa = kilo Dalton; µm = Micrometer. Bars = 10 µm deleted. Line 249 -250.

  1. [Page 10, lines 319-330] It is recommended to delete the whole section 2.8.

Response: Section removed.

  1. [Page 12, lines 420-422] Please delete the following sentence: The study was performed with strict adherence to the guidelines and principles ensuring participants’ safety.

Response: Statement deleted. Line 397.

Reviewer 2 Report

Comments and Suggestions for Authors

The manuscript addresses the issue of genetic identification of HI in African subregions. By combining whole exome sequencing, heterologous mutant expression, western blotting and cell fluoresce studies the authors made robust findings of a novel variant in the MARVELD2 gene. The mutation segregated with the clinical phenotype and was corroborated by rigorously conducted functional studies. The methodologies are state-of-this art and well conducted, results well interpreted and discussion straightforward. The identification of gene variants in rare and ultrarare conditions in Africa and African subregions is a major issue in medical genetics and the findinsg of the authors increase our knowledge on this topic. 

Author Response

Many thanks for your excellent summary of our work and appreciative words.

Round 2

Reviewer 1 Report

Comments and Suggestions for Authors

It seems that the revised version was properly edited for the comments.